# Trusting the experts: The domain-specificity of prestige-biased social learning

**Charlotte O. Brand**[1,2]*, **Alex Mesoudi**[1], **Thomas J. H. Morgan**[3,4]

**1** Department of Biosciences, Human Behaviour and Cultural Evolution Group, College of Life & Environmental Sciences, University of Exeter, Exeter, England, **2** Department of Psychology, University of Sheffield, Sheffield, England, **3** School of Human Evolution and Social Change, Arizona State University, Tempe, AZ, United States of America, **4** Institute of Human Origins, Arizona State University, Tempe, AZ, United States of America

\* c.brand@sheffield.ac.uk

**Data Availability Statement:** The data underlying the results presented in the study are available from https://github.com/lottybrand/Prestige_2_Analysis.

## Abstract

Prestige-biased social learning (henceforth "prestige-bias") occurs when individuals predominantly choose to learn from a prestigious member of their group, i.e. someone who has gained attention, respect and admiration for their success in some domain. Prestige-bias is proposed as an adaptive social-learning strategy as it provides a short-cut to identifying successful group members, without having to assess each person's success individually. Previous work has documented prestige-bias and verified that it is used adaptively. However, the domain-specificity and generality of prestige-bias has not yet been explicitly addressed experimentally. By domain-specific prestige-bias we mean that individuals choose to learn from a prestigious model only within the domain of expertise in which the model acquired their prestige. By domain-general prestige-bias we mean that individuals choose to learn from prestigious models in general, regardless of the domain in which their prestige was earned. To distinguish between domain specific and domain general prestige we ran an online experiment (n = 397) in which participants could copy each other to score points on a general-knowledge quiz with varying topics (domains). Prestige in our task was an emergent property of participants' copying behaviour. We found participants overwhelmingly preferred domain-specific (same topic) prestige cues to domain-general (across topic) prestige cues. However, when only domain-general or cross-domain (different topic) cues were available, participants overwhelmingly favoured domain-general cues. Finally, when given the choice between cross-domain prestige cues and randomly generated Player IDs, participants favoured cross-domain prestige cues. These results suggest participants were sensitive to the source of prestige, and that they preferred domain-specific cues even though these cues were based on fewer samples (being calculated from one topic) than the domain-general cues (being calculated from all topics). We suggest that the extent to which people employ a domain-specific or domain-general prestige-bias may depend on their experience and understanding of the relationships between domains.

**Funding:** CB and AM were supported by The Leverhulme Trust (grant no. RPG-2016-122 -awarded to AM) https://www.leverhulme.ac.uk/ The funders had no role in study design, data collection and analysis, decision to publish, or preparation of the manuscript.

**Competing interests:** The authors have declared that no competing interests exist.

## Introduction

The field of cultural evolution seeks to explain broad patterns of cultural change and variation in terms of, amongst other factors, the means by which information is passed from person to person via 'social learning biases' [1–5]. Here, 'bias' is meant in the statistical sense of a systematic departure from randomly choosing another person from whom to learn (rather than the normative sense of 'bias', implying error or mistakes). Examples of such social learning biases include copying the majority (conformity), copying older individuals, copying when uncertain, or copying prestigious individuals [6]. In this study we focus on the latter: prestige-biased social learning (henceforth prestige-bias), which occurs when learners preferentially learn from individuals who are copied by others, attended to by others, or who generally receive freely conferred deference from others [7–14]. These individuals are said to have 'prestige'.

Prestige-bias has been proposed as an adaptive social learning strategy as it provides an efficient short-cut to acquiring adaptive social information when more direct means (e.g. identifying and copying the most skillful/knowledgeable individuals) are unavailable or costly [11]. According to the cultural evolutionary theory of prestige, prestige-bias is only adaptive *because* the prestige was first acquired due to success, such that prestige-cues (e.g. being copied by, or receiving attention from others) are indirect proxies for success [11]. Consistent with this theory, our previous experimental work shows that participants use prestige information (e.g. who others are copying) when selecting a model from whom to socially learn, but only when a) the prestige information is to some extent related to success and b) direct success information (e.g. score) is unavailable [7]. Importantly, such conclusions were obtained when prestige was an emergent property of participants' behaviour during the experiment; no deception or manipulation of prestige was employed at any time, thus providing a naturalistic test of the prestige bias theory [7].

Here we build on our previous study [7] by exploring the domain-specificity or domain-generality of prestige-bias, which remains poorly understood. By the terms 'domain-specificity' and 'domain-generality' we mean whether a prestigious individual's influence is limited to the specific domain in which they are successful (domain-specific prestige) or whether prestigious individuals who are successful in one domain become influential even outside their domain of expertise (domain-general prestige). For example, extremely prestigious individuals who have gained fame for their expertise are often used as a source of advice outside the domain in which they gained expertise, such as famous sportspeople giving political or medical advice.

Consequently, we can ask whether social learners specifically copy individuals who are prestigious in the domain of interest (domain-specific prestige-bias), or if they copy individuals who are prestigious regardless of the source of this prestige (domain-general prestige-bias). For instance, is a successful and prestigious boat-builder sought out for advice on all kinds of general matters, or is their influence limited to their specific craft? In the original specification of prestige bias, Henrich & Gil-White, p.170 [11] suggest that ". . .prestige hierarchies can be domain-specific. For example, if I defer to you because of your superior computer skills and you defer to Bob because he is an excellent grass hockey player, I may not give Bob any special deference if grass hockey is not my thing," [11]. However, they also go on to suggest that "prestigious individuals are influential even beyond their domain of expertise," p.184 [11], and it is often claimed in the literature that prestigious celebrities who have acquired fame via a specific domain, such as sport or acting, have influence beyond their domain of expertise (see [15] for examples and critical discussion). As such, the domain-specificity of prestige-bias remains unclear.

We suggest that domain-general prestige-bias will be adaptive when success across different domains is correlated. As noted by Henrich & Gil-White, "much of the information that leads to success in one domain will often be transferable to others. . . acquiring skill in one domain (e.g. a martial art) is often touted as promoting success in many other areas. For example, problem-solving methods, goal-achieving strategies, eye-hand coordination, control over one's emotions, etc. are useful across several domains," [11]. Henrich & Gil-White go on to suggest that "because figuring out which combinations of ideas, beliefs, and behaviours make someone successful is costly and difficult, selection favoured a general copying bias, which also tends to make prestigious individuals generally influential (as people copy and internalise their opinions)." (p.184). We suggest that the domain-generality of prestige-bias is an open empirical question, and is dependent on not only how correlated the domains in question are, but also whether observers have an accurate perception of these correlations.

Previous research shows some evidence of the domain-specificity of prestige-bias. One experimental study found that 3–4 year old children preferentially copied the artefact choice of a prestigious demonstrator, defined as the demonstrator to whom bystanders had attended rather than ignored, when their prestige was acquired during an artefact task (i.e. in the same domain), but not when it was acquired on a food-preference task (i.e. in a different domain) [10]. The reverse preference was seen when the demonstrator acquired their prestige on the food-preference task rather than the artefact choice task. Other studies from naturally-occurring groups have provided indirect evidence that prestige-bias may not be as domain-general as suggested by Henrich & Gil-White. For example, one study found that prestige was unrelated to ethnobotanical medicinal knowledge in an indigenous Tsimane community, but instead was related to having a formal position in that society [16]. Their measure of prestige was the number of nominations received when others were asked to list all the important men in the village, suggesting that being highly skilled and knowledgeable in the domain of medicine does not translate to being an important man in the village generally, or gaining a formal position. Similarly, a study of naturally-occurring volunteer groups in Cornwall, such as chess or kayaking clubs, found that prestige ratings were not related to success in a quiz that the group completed together, but *were* related to formal positions that the individuals already had in their groups, such as teacher, team captain, or group organiser/secretary [17].

Here we present an experiment to directly assess the degree to which participants use domain-specific or domain-general prestige cues when choosing from whom to learn, building on our previous study [7]. As before, instead of manipulating prestige experimentally, we use participant-generated prestige cues and manipulate the choices participants face to directly compare the different types of cues, providing a suitable combination of naturalistic prestige-generation yet also experimental manipulation of our key variable of interest (domain-specificity). Participants answered quiz questions from four different topics: weight estimation, world geography, art history, and language identification. We treat each of these topics as a 'domain', and ask whether people use success-derived prestige in one domain/topic as a guide for who to copy in a different domain/topic.

On each of 100 questions, participants could answer for themselves or copy another individual (henceforth 'demonstrator') from their group. Depending on the experimental condition, participants could choose who to copy based on (1) the demonstrators' scores on questions from the current topic (domain-specific success) or (2) the number of times the demonstrators were previously copied (our experimental proxy for prestige) on (a) questions from the current topic (domain-specific prestige), (b) all topics (domain-general prestige) or (c) a different topic (cross-domain prestige).

Our previous experiment using the same quiz and topics [7] found that participants' scores across both rounds of questions from the same topic were more strongly associated (r between

0.55 and 0.78), than scores across both rounds of different topics (r between 0.24 and 0.6). The associations between scores on a particular topic and overall score across rounds were intermediate between these two ranges. These associations therefore indicate that domain-specific scores (scores on questions within the same topic) are the most reliable cue to choose potential demonstrators from whom to copy when answering questions from the same topic/domain, compared to cross-domain or domain-general cues. Participants, however, were not informed of these correlations, nor had any way to calculate them during the experiment.

Given that domain-specific prestige (derived from the same topic) should be most highly correlated with score on the topic being answered, and that cross-domain prestige (derived from a different topic) will be least correlated with score on the topic being answered, and domain-general prestige bias (derived from all topics) will be intermediate, we formulated the following hypotheses, preregistered at https://osf.io/93az8.

1. Domain-specific prestige-bias will be employed more often than domain-general prestige-bias and cross-domain prestige

2. Domain-general prestige-bias will be employed more often than cross-domain prestige-bias

3. Cross-domain prestige-bias is only employed when access to domain-general or domain-specific prestige information is not available.

## Methods

This research was granted ethical approval by the University of Exeter Biosciences ethics committee. Approval number: eCORN001806 v8.1.

### Participants

Consistent with our previous study [7], we aimed to recruit ten groups of ten participants via Mechanical Turk for each of four conditions, totaling 400 participants. Participants were randomly assigned to one of ten groups within one of four conditions, giving a between-participant design. Due to difficulties in coordinating multiple groups of participants simultaneously, some participants dropped out at various stages, giving a starting sample size of 397, with 332 participants reaching the final demographics page. We ensured a minimum of 5 participants in each networked group throughout. All participants were above the age of 18 (age range 20–69, mean age = 39.1), with 113 men and 78 women (not every participant specified their gender). All were given a monetary reward for their time of USD$10, and had the opportunity of winning a bonus payment of $20 if they scored over 85% in the quiz. 33 participants received the $20 bonus payment. All participants provided informed consent before being able to proceed to the task. The consent form stated that their participation was entirely voluntary, their data were entirely anonymous, and they could withdraw their involvement at any time by closing their browser.

### Materials

The experimental automation platform Dallinger (dallinger.readthedocs.io) was used to create an online game in which groups of players can play and interact simultaneously. Participants answered 100 questions with two alternative answers each, one correct and one incorrect. The 100 questions were split into four topics of 25 questions each: geography, weight estimation, language, and art (see Fig 1 for an example).

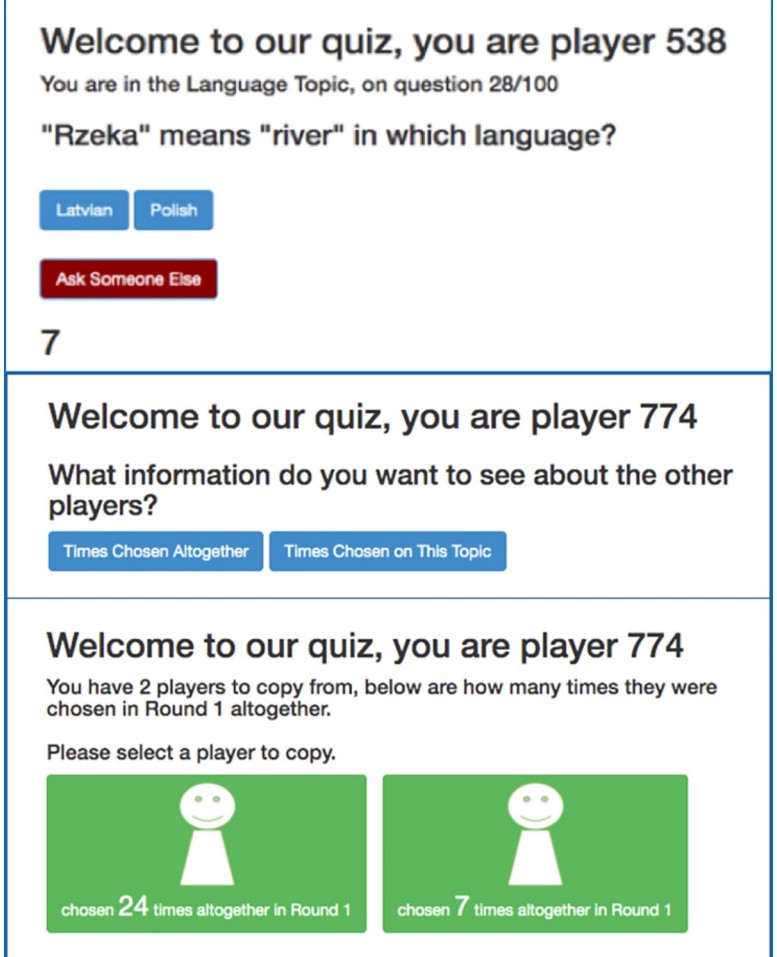

**Fig 1. Three example screenshots representing what participants saw at different stages of the experiment.** The top screenshot is an example question from the language topic. Participants could either select one of the two blue buttons showing two possible answers (one correct, one incorrect), or select the red button labelled "Ask Someone Else" which allows participants to copy someone else within their group. The number '7' at the bottom is a countdown timer that forces participants to answer within 15 seconds. The second image represents what a participant would see if they chose to "Ask Someone Else" in Round 2 of Condition C, where they could choose to either view Times Chosen Altogether (domain-general prestige) or Times Chosen On This Topic (domain-specific prestige). The bottom image represents what a participant would see if they chose 'Times Chosen Altogether' and (domain-general prestige), in which there were only two other players to choose from. Please note that for any given question, participants could have between one and nine other participants to choose from, depending on how many answered individually for that particular question. See Table 1 for the information combinations displayed in the other Conditions. See S1 File for screenshots for all Conditions.

## Procedure

Participants were given 100 binary choice questions based on four different general-knowledge or 'trivia' style topics, 25 in each category. Participants had fifteen seconds to answer each question, and they scored one point for every question they answered correctly. On each question, instead of answering themselves, participants could choose to "Ask Someone Else" by clicking the corresponding button. This allowed them to see information about the other participants ('demonstrators') in their group who had answered that question themselves (if everyone chose to "Ask Someone Else" on a single question then participants were shown a message saying "sorry, everyone chose to 'ask someone else' so no one can score points for this

**Table 1. Information displayed when choosing to "Ask Someone Else" in Rounds 1 and 2 across conditions, with our predicted choice for Round 2 in bold.**

|  | **Specific v Cross** (Condition A) | **General v Cross** (Condition B) | **Specific v General** (Condition C) | **Cross v Random** (Condition D*) |
|---|---|---|---|---|
| Round 1 | Domain-specific score | | | |
| Round 2 | **Domain-specific prestige** Or Cross-domain prestige | **Domain-general prestige** Or Cross-domain prestige | **Domain specific prestige** or Domain-general prestige | **Cross-domain prestige** Or Random cue |

Specific v Cross (Condition A) compares domain-specific and cross-domain prestige; General v Cross (Condition B) compares domain-general and cross-domain prestige; Specific v General (Condition C) compares domain-specific and domain-general prestige; and Cross v Random (Condition D*) compares cross-domain prestige with a random cue entirely unconnected to success in the task. Please note Condition D was run separately after confirmatory analysis on Conditions A- C, as specified on p.7 of our Pre-registration.

question" because there would be no answers available to copy for that question). The information they saw depended on the condition, detailed in Table 1. They then chose a demonstrator whose answer they used for that question. If the chosen demonstrator answered the question correctly, the copying participant also scored a point for that question. If the demonstrator was incorrect, they did not. No one received feedback on whether their answer was right or wrong at any point.

In all conditions, the 100 questions were split into two rounds, 60 questions in Round 1 (15 from each topic), and 40 in Round 2 (10 from each topic). In Round 1, when participants chose to "Ask Someone Else", they always saw the current domain-specific score of all available demonstrators. However, in Round 2, participants could choose between two kinds of information, which varied according to the experimental condition (see Table 1). The information available included domain-specific prestige (the number of times each demonstrator was copied on Round 1 questions from this topic), domain-general prestige (the number of times copied on all Round 1 topics), cross-domain prestige (the number of times copied on Round 1 questions from a randomly selected, different topic) and a random cue (each demonstrator's Player ID, which was a randomly generated 3 digit number). The four conditions pitted different pairs of information against each other, to establish participants' preferences, and across the four conditions we constructed a hierarchy of information, from most to least preferred.

## Pre-registered predictions and analyses

The following sets of predictions were preregistered at https://osf.io/93az8. Predictions 1–3 are assumption checks, to make sure that the correlations within and across domains were suitable for testing the subsequent predictions. Predictions 4–6 are our main theoretically-derived, *a priori* predictions shown in Table 1. Predictions 7 and 8 are follow-up predictions looking at copying frequency and score across conditions. Finally, we have predictions related to our preregistered follow-up Condition, (Condition D), which was run after analysing results for Conditions A-C, as specified in the pre-registration (p.7 heading "Follow-up Analyses"). Please note that Assumption Check 1 differs to our preregistration. This was an oversight, as we wanted to check the assumption that participant *scores* were more tightly correlated within compared to between topics, not participant *prestige*, in the same way that we did for our previous study (Brand et al. 2020).

To analyse our data we ran a series of Bayesian multi-level mixed models using the Rethinking package version 1.90 [18] in R version 3.6.0 [19]. Our raw data and analysis scripts can be found at https://github.com/lottybrand/Prestige_2_Analysis. Each model corresponding to each prediction is also included in the S1 File. Model parameters are interpreted as providing evidence of an effect on the outcome if their 89% credible interval (CI) did not cross zero, with

89% CIs the default value in the Rethinking package. 89% credible intervals are used to prevent readers from confusing these with the widely-used 95% confidence intervals and performing unconscious significance tests [18]. Priors were chosen to be weakly regularising, in order to control for both under and overfitting the model to the data. Convergence criteria such as effective sample sizes and Rhat values were used to check for appropriate model convergence throughout, and trace plots were inspected for signs of incomplete mixing when necessary.

## Predictions 1–3 (assumption checks)

1. Domain-specific scores (i.e. the number of times a participant scored correctly in each topic) are more highly correlated between rounds than domain-general score or cross-domain score. This is an assumption check based on our previous findings and the experimental set-up. If this prediction is not upheld it suggests the topics were more tightly correlated than anticipated, and our following predictions may not hold. To test Prediction 1 we calculated Pearson's Correlation Coefficients between participant's asocial scores in Round 1 and Round 2 within each topic, and compared these to correlation coefficients between different topics, and compared to overall score.

2. Across all conditions, when choosing to "Ask Someone Else" in Round 1, participants preferentially copy the highest-scoring demonstrator. This is an assumption-check, to make sure that subsequent copying frequency cues are genuine signals of performance, and a replication of our previous findings. To test this prediction we scored each trial for whether or not the participant chose to copy the highest scoring demonstrator available for each copying instance, and used this as the outcome variable in a binomial model with varying intercepts for group and participant. (Model 1 in S1 File).

3. Across all conditions, when choosing to "Ask Someone Else" in Round 2, participants preferentially copy the most-copied demonstrator. This is an assumption-check to make sure that people are actually employing prestige-bias when it is potentially useful, and a replication of our previous findings. To test this prediction we scored each trial for whether or not the participant chose to copy the most copied demonstrator available for each copying instance, and modelled this as above (Model 2 in S1 File).

## Predictions 4–6 (main A-priori predictions)

4. In the Specific v Cross Condition (A), participants will predominantly choose to copy based on domain-specific prestige, rather than cross-domain prestige, as domain-specific prestige will be most correlated to score on the relevant domain.

5. In the General v Cross Condition (B), participants will predominantly choose to copy based on domain-general prestige, rather than cross-domain prestige, as domain-general prestige will be more correlated to score on the relevant domain than cross-domain prestige (as it contains the topic-specific score within it).

6. In the Specific v General Condition (C), participants will predominantly choose to copy based on domain-specific prestige rather than domain-general prestige, as domain-specific prestige will be more correlated to score on the relevant domain.

To test Predictions 4, 5 and 6 a generalised linear mixed model (Model 3 in S1 File) of all data from Round 2 was used with "chose predicted" (i.e. chose to view the information that we

predict in each condition) as the binomial outcome variable (yes or no), with varying intercepts for participant, group and condition.

### Predictions 7–8 (follow-up predictions)

7. The overall frequency of copying (i.e. choosing to "Ask Someone Else") is higher in the Specific v Cross Condition (A) and the Specific v General Condition (C) than in the General v Cross Condition (B). This is because Conditions A and C both provide domain-specific prestige information which is a more direct cue to domain-specific success, whereas Condition B provides only indirect cues to domain-specific score. To test this prediction, a generalized linear mixed model of all data from Round 2 was used with 'copied' as the binomial outcome variable (i.e. chose "Ask Someone Else" or not)

8. Similarly to above, participants score higher in Conditions in which domain-specific prestige is available because they provide more tightly correlated cues to success with the relevant domain. To test this prediction, a general linear mixed model of data from each condition was used with final score of each participant as the outcome variable and condition as the predictor variable, with a varying intercept for participant and group.

### Unregistered predictions

As laid out in the preregistration (p.7 under heading "Follow-up Analyses"), after collecting and analysing data for Conditions A-C and confirming that participants showed a clear preference for domain-specific information over cross-domain and domain-general prestige, we then collected additional data for Condition D which compared cross-domain prestige (the least preferred information) with a random cue (Player ID number). Thus for Condition D, we predicted that participants would predominantly choose to copy based on cross-domain prestige rather than the random cue, as cross-domain prestige contains some correlation with score, whereas the random cue does not. Because Condition D contains the least favoured information, we also predicted lower copying rates and lower scores than the other conditions.

## Results

### Assumption check (Prediction 1)

As predicted, scores were more strongly associated between Rounds 1 and 2 of the same topic than they were between different topics (Table 2), except for the Geography topic in which the within-topic correlation (r = .47) was lower than two cross-topic correlations (Art (r = .55) and Language (r = .51)). Also as predicted, the overall Round 1 score was intermediate between the within-topic associations and the cross-topic associations for Art and Weight, but not for Geography and Language where the overall correlation was higher than the within-topic correlation. This latter unexpected finding may reflect the fact that overall score contains four times as much data as does the score for any specific topic.

### Assumption checks (Predictions 2 and 3)

As predicted, when participants chose to copy others in Round 1 they preferentially copied the most successful (i.e. highest scoring) demonstrator out of all those available (mean intercept estimate: 3.15, 89% CI: 2.77, 3.55, this corresponds to participants choosing the highest scoring model 95.9% of the time on the probability scale). When participants chose to copy in Round 2, participants preferentially copied the most prestigious (i.e. most copied) demonstrator out

**Table 2. Correlation coefficients between Round 1 and Round 2 topic scores representing total variance of Round 2 topic score explained by Round 1 score of the same topic.**

| Round 1 asocial score | Round 2 asocial score | Correlation coefficient (95% confidence intervals) |
|---|---|---|
| **Art** | **Art** | **0.71 (0.66, 0.76)** |
| Geography | Art | 0.43 (0.43, 0.52) |
| Language | Art | 0.50 (0.41, 0.58) |
| Weight | Art | 0.38 (0.29, 0.47) |
| Overall Round 1 Score | Art | 0.64 (0.57, 0.70) |
| **Geography** | **Geography** | 0.47 (0.38, 0.55) |
| Art | Geography | 0.55 (0.47, 0.62) |
| Language | Geography | 0.51 (0.42, 0.58) |
| Weight | Geography | 0.38 (0.29, 0.47) |
| Overall Round 1 Score | Geography | **0.59 (0.52, 0.66)** |
| **Language** | **Language** | 0.59 (0.51, 0.65) |
| Art | Language | 0.54 (0.46, 0.62) |
| Geography | Language | 0.42 (0.33, 0.51) |
| Weight | Language | 0.37 (0.28, 0.46) |
| Overall Round 1 Score | Language | **0.60 (0.53, 0.67)** |
| **Weight** | **Weight** | **0.57 (0.49, 0.63)** |
| Art | Weight | 0.47 (0.39, 0.55) |
| Geography | Weight | 0.29 (0.19, 0.39) |
| Language | Weight | 0.42 (0.33, 0.51) |
| Overall Round 1 Score | Weight | 0.55 (0.47, 0.62) |

All scores are participants' *asocial* score, so only includes scores they achieved without copying others. *Bold* coefficients shows highest association for that topic.

of all those available (mean intercept estimate: 4.29, 89% CI: 3.59, 5.10, this corresponds to participants choosing the highest scoring model 98.6% of the time on the probability scale).

## A-priori hypothesis tests (Predictions 4, 5 and 6)

As shown in the raw data presented in Fig 2, participants preferred the domain-specific prestige cue when available, followed by domain-general, then cross-domain, and finally random cues, as predicted. Our models confirmed that participants preferentially chose the predicted information as opposed to the alternative information in each condition we tested, showing a strong preference for the predicted information (see Fig 3; **Condition A**: mean: 3.05 89% CI: 2.42, 3.73 **Condition B**: 2.51, 89% CI: 1.90, 3.16, **Condition C**: mean: 2.84, 89% CI: 2.24, 3.46 **Condition D**: 2.10, 89% CI: 1.34, 2.87).

## Follow-up analyses (Prediction 7)

Our model did not provide strong evidence of a difference in copying frequency between Conditions A (18% copying rate, n = 3240), B (14% copying rate, n = 3437) and C (20% copying rate, n = 3377), however there is evidence of participants copying less in Condition D compared to the other conditions (7% copying rate, n = 3416). This was confirmed by computing contrasts between the estimates of the four conditions (mean difference between Conditions A and D: 1.77, 89% CI: 0.72, 2.81. Mean difference between Conditions B and D: 1.26, 89% CI: 0.29, 2.24. Mean difference between Conditions C and D: 1.83, 89% CI: 0.76, 2.86).

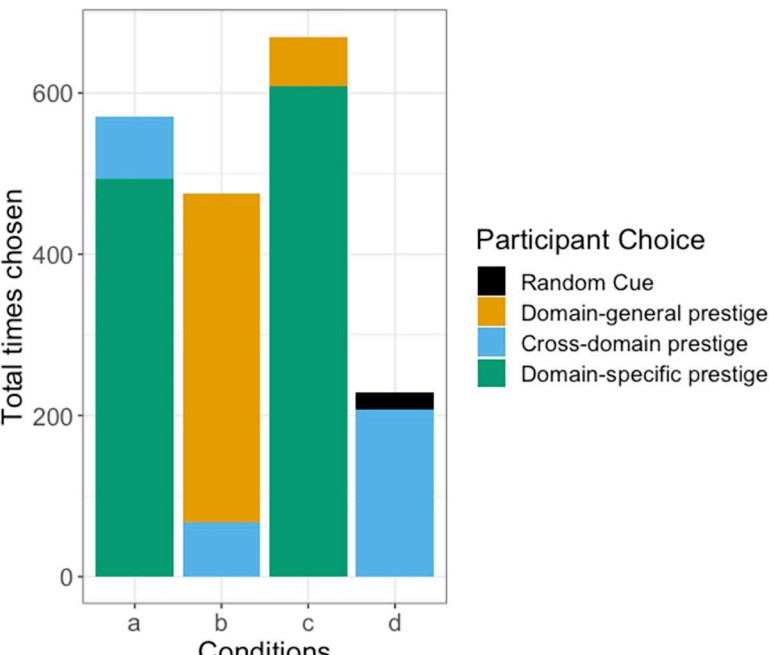

**Fig 2. Raw counts of the information chosen when participants chose to copy someone else's answer in Round 2.**
Total possible copying instances for each condition in Round 2 were: Condition A = 3240, Condition B = 3437,
Condition C = 3377, Condition D = 3416.

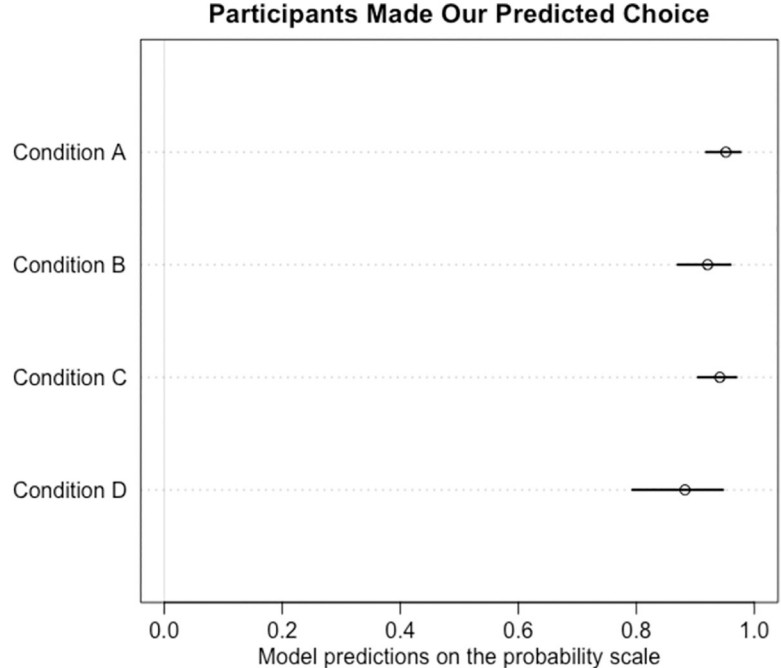

**Fig 3. Model predictions for participants choosing the predicted information compared to the alternative
information in Round 2 of the four conditions, on the probability scale.**

### Follow-up analysis (Prediction 8)

Similarly to prediction 7, there was not strong evidence that participants scored differently between Conditions A (mean score 71.1%), B (71.6%) and C (72.5%). However, there was strong evidence that participants scored lower in Condition D (mean score 63.5%) compared to Condition C only. This was confirmed by computing contrasts between the conditions (mean difference between Condition D and C: -1.15, 89% CI: -2.23, 0.04). When looking at just the Round 2 scores (in which there was a difference in the information participants could choose), participants scored lower in Condition D (mean score 24.8) compared to Condition B (mean score 27.4, mean difference -1.18 89% CI: -2.02–0.33) and Condition C (mean score 27.4, mean difference -1.19 89% CI: -2.03, -0.32), but not Condition A (mean score 26.5).

The S1 File contains model specifications, a comparison with our previous prestige study that used a similar design [7] and further exploratory analyses, including results showing that copying rate predicts score, and plots showing individual variation in prestige score and copying rate.

## Discussion

In order to acquire adaptive information, people often preferentially learn from prestigious individuals, where 'prestige' is acquired by being observed and copied by others. While this prestige-biased social learning has been demonstrated in previous research [7], uncertainty remains over the extent to which prestige bias is domain-specific, where people preferentially copy others who have acquired prestige in the same domain as what is being copied, or domain-general, where people preferentially copy others who are generally copied on a range of domains. In this study we experimentally tested whether participants adopted a domain-specific prestige-bias or a domain-general prestige-bias when both options were available to them during an online quiz. This quiz contained different topics that represented different 'domains' of knowledge. Participants were able to copy other players' quiz answers based on their domain/topic-specific scores in Round 1 of the quiz, and were then able to copy other players based on how many times they had previously been copied by others (our measure of 'prestige') in Round 2 of the quiz. In a series of pair-wise comparisons between prestige cue types, we found that participants overwhelmingly chose to use domain-specific prestige cues (times copied in the same topic) over both domain-general (time copied overall) and cross-domain (times copied in a different randomly chosen topic) prestige cues, that they preferred domain-general to cross-domain cues, and that they preferred cross-domain to a random cue entirely unconnected to success in the task. We therefore revealed a 'hierarchy' of prestige cues in which the most favoured cue, when available, was domain-specific prestige, followed by domain-general, then cross-domain, and lastly random cues. Importantly, as with our previous work [7], prestige cues were an emergent property of participants' behaviour during the experiment; no deception or manipulation of prestige cues was employed at any time, thus increasing our confidence that such effects might be observed in the real world.

This study adds to the already extensive body of work which shows that people tend to use social information in an adaptive and flexible way, depending on the information that is available to them [5,6,20]. We also provide further evidence of success-biased social learning, in that participants copied the highest scoring player available to them when copying based on score in Round 1 [5,7,21]. Participants also copied the most-copied player available to them when copying based on prestige in Round 2. Our results also support previous evidence suggesting that participants are sensitive to the domain-specificity of prestige when copying or learning from others [10,16,17].

Our study used different topics of a quiz to represent different domains of knowledge, and participants' scores in Round 1 of a given topic were generally better predictors of their score in Round 2 of the same topic than of any other topic. Thus, copying based on domain-specific prestige was most adaptive for increasing a player's chances of selecting the correct answer on any given question, and ultimately in achieving a high score (and bonus payment) in the quiz. As the quiz was played live by groups of participants with no feedback on their scores, participants did not have access to how correlated scores were within or between topics. Participants did have experience of each topic in Round 1, presumably aiding their assessment of how related the topics were. Similarly, in real life people do not have direct access to data on how correlated different domains of skill or knowledge are, but presumably gain an intuitive understanding based on their experience or exposure to different domains during their lifetimes. Our predictions assume that people have an intuitive sense of which domains should be correlated, based on their experience. Thus, an individual's tendency to prefer domain-specificity over generality relies on their own understanding or assessment of how correlated the domains might be. How accurate their perceptions of the correlations are depends on their own experience and expertise in those domains, and likely emerges either through unconscious associative learning, conscious deliberation, or a combination of the two. If direct experience of a domain is lacking, intuitions might be based on stereotypes or folk understanding, leading to inaccurate assessment of relationships between domains, and maladaptive use of prestige-bias. We feel our results are reassuring, in that they suggest people are sensitive to the source of people's prestige, and that if access to a more relevant source is provided, they will preferentially use it over a less accurate source.

We found strong evidence that participants prefer domain-specific prestige cues to cross-domain prestige-cues, i.e. they would rather copy the answer of someone who was copied extensively on the current topic, than someone who was copied extensively on a different topic. Our topics were explicitly labelled as different to each other, and participants had experience of each topic in Round 1. This explicit labelling and direct experience may be responsible for participants' strong domain-specific bias. However, if topics were more alike, participants may not show such a strong preference for domain-specific prestige. Importantly, if they had less experience of the topics or if the topics were unfamiliar to them, they may be less able to make a judgement about their correlations. For example, an individual unfamiliar with science may see both a physicist and a biologist as belonging to the same domain of expertise, 'scientist', and perhaps incorrectly choose to learn about physics from a biologist. The biologist may be a better model compared to an historian or an artist, but another scientist might choose the more domain-specific model, i.e. the physicist. This may help to explain why prestigious individuals often influence others on topics outside of their domain of expertise, especially within celebrity culture and social media platforms such as Twitter. It may be more informative for someone to listen to a prestigious children's author on a topic related to biology if they have no other access to biology experts. However another biologist may choose not to listen to a children's author on this topic as they have access to plenty of other biologist models. Perhaps if people had wider and more direct access to experts on particular topics, they would preferentially weight the opinion of the expert in their domain of expertise rather than someone who had gained expertise in an unrelated different topic. An anecdotal example of this has been demonstrated during the Covid-19 pandemic, in which many scientists have started following epidemiologists on Twitter in order to gain the most domain-specific information relating to the pandemic. However, those outside of science may view any Biologist as equally prestigious, despite having no epidemiological expertise.

As noted by Henrich & Gil-White [11], the emergence of domain-general prestige depends on whether success in multiple domains depends on a common underlying trait. In our quiz,

the topics were chosen to be as dissimilar to each other as possible (within the constraints of an online quiz), however we showed that performance across topics was to some extent correlated, either because the domains are drawing on similar cognitive abilities, or because success across domains is correlated with an underlying trait such as education, socioeconomic status, or 'general intelligence' such as 'g' [22–25]. The theoretical justifications and debates surrounding the construct of *g* are beyond the scope of this paper, but if theoretical and statistical models of *g* are reliable and replicable, then this would support the argument that domain-general prestige-bias can often be as adaptive as, if not more adaptive than, domain-specific prestige. Just as a general intelligence factor may underpin general cognitive abilities, success across many broad areas of societal interest, such as politics, science or ethics, could plausibly be correlated with underlying traits such as reasoning, decision-making, or evidence-based judgement. For this reason it could be adaptive to use domain-general prestige-bias for large-scale decisions. For example, for successful decision-making on climate change policy, an understanding of data, evidence gathering, political landscapes and policy logistics may all be necessary, and may all be reliant on common training, expertise or experience. In this way, a renowned "thinker" or "intellectual" (e.g. a broad-scope podcast host such as Sean Carroll) may be just as informative on climate policy as a climate scientist; the climate scientist may know the climate change data exceptionally well but may not have any experience of the political landscape or policy logistics.

Interestingly, domain-specific prestige was overwhelmingly preferred to domain-general prestige, even though in our experimental setup the former actually contained a lower volume of information, being based on 15 topic-specific questions from Round 1, than the latter, which was based on all 60 questions from Round 1 (and indeed overall score in Round 1 was a better predictor of topic specific score in Round 2 in two out of four topics). Thus the domain-general prestige information included copying instances from four times as many questions compared to the domain-specific prestige information. This difference in information was also reflected in the prestige scores visible to participants, in that more copying instances would have occurred across all topics compared to within each topic. This suggests that people might be less sensitive to the amount of information than might be expected. Whether our experimental set-up reflects real life depends on a trade-off between depth and breadth of expertise. For example, someone may have a long history of prestige in biology generally, thus lots of information has been gathered about their success as a biologist. Someone else may have recently gained expertise in a specific area of biology, such as molecular genetics. Our results suggest people would rather learn molecular genetics from the recently trained molecular genetics expert than the long-standing general biology expert. Similarly, if someone is renowned for being generally successful or knowledgeable in a variety of domains, do we have more information about their expertise than about the expertise of someone who is a long-time expert in a narrower field? This trade-off between depth and breadth of expertise is an open-question and one worth investigating when trying to understand who people choose to learn from and why.

In summary, we find evidence that domain-specific prestige-bias is preferred to both cross-domain and domain-general prestige-bias, at least when the domains of interest are sufficiently dissimilar to each other, and when individuals have had experience of each domain. Participants revealed a hierarchy of prestige in that domain-specific prestige was most preferred, followed by domain-general, and cross-domain prestige was least preferred. This preference was present despite the fact that the domain-general prestige scores contained more information (were generated based on all topics) compared to domain-specific prestige scores (generated from one topic). This apparent trading of depth over breadth of expertise warrants further experimental investigation to understand who people learn from and why.

## Supporting information

**S1 File.**
(PDF)

## Author Contributions

**Conceptualization:** Charlotte O. Brand, Alex Mesoudi, Thomas J. H. Morgan.

**Data curation:** Charlotte O. Brand, Thomas J. H. Morgan.

**Formal analysis:** Charlotte O. Brand.

**Funding acquisition:** Alex Mesoudi.

**Methodology:** Charlotte O. Brand, Thomas J. H. Morgan.

**Project administration:** Alex Mesoudi.

**Resources:** Alex Mesoudi.

**Software:** Thomas J. H. Morgan.

**Supervision:** Alex Mesoudi, Thomas J. H. Morgan.

**Validation:** Alex Mesoudi.

**Visualization:** Charlotte O. Brand.

**Writing – original draft:** Charlotte O. Brand.

**Writing – review & editing:** Charlotte O. Brand, Alex Mesoudi, Thomas J. H. Morgan.

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
