## [Decision Letter · Decision Letter 0]

10 Feb 2021

PONE-D-20-39514

Trusting the experts: the domain-specificity of prestige-biased social learning

PLOS ONE

Dear Dr. Brand,

Thank you for submitting your manuscript to PLOS ONE. After careful consideration, we feel that it has merit but does not fully meet PLOS ONE’s publication criteria as it currently stands. Therefore, we invite you to submit a revised version of the manuscript that addresses the points raised during the review process.

As you will see, all the reviewers see merit in your study and provide overly positive reviews. Reviewer 3 recommends accepting the present version as is. Reviewers 1 and 2 in contrast suggest a number of revisions. In particular, both reviewers think that the the readability of the paper can be improved, particularly in the Methods and Results sections. I share their concerns (it was also difficult for me to understand those sections of the paper on a first read) and encourage you to take their suggestions to the heart. In addition to the detailed analytical recommendations made by Reviewer 1, I'd suggest reporting raw correlations (instead of R2) as a measure of association between variables, because they are substantially easier to interpret for most readers (R2 are absolutely fine as a measure of model fit, but I think they can be misleading in this context).

We look forward to receiving your revised manuscript.

Kind regards,

Miguel A. Vadillo, Ph.D.

Academic Editor

PLOS ONE

Journal Requirements:

2.  We note that Figure 1 in your submission contain map images which may be copyrighted. All PLOS content is published under the Creative Commons Attribution License (CC BY 4.0), which means that the manuscript, images, and Supporting Information files will be freely available online, and any third party is permitted to access, download, copy, distribute, and use these materials in any way, even commercially, with proper attribution. For these reasons, we cannot publish previously copyrighted maps or satellite images created using proprietary data, such as Google software (Google Maps, Street View, and Earth). For more information, see our copyright guidelines: http://journals.plos.org/plosone/s/licenses-and-copyright.

2.1.    You may seek permission from the original copyright holder of Figure 1 to publish the content specifically under the CC BY 4.0 license. 

2.2.    If you are unable to obtain permission from the original copyright holder to publish these figures under the CC BY 4.0 license or if the copyright holder’s requirements are incompatible with the CC BY 4.0 license, please either i) remove the figure or ii) supply a replacement figure that complies with the CC BY 4.0 license. Please check copyright information on all replacement figures and update the figure caption with source information. If applicable, please specify in the figure caption text when a figure is similar but not identical to the original image and is therefore for illustrative purposes only.

3. We note that Figure 1 in your submission contain copyrighted images. All PLOS content is published under the Creative Commons Attribution License (CC BY 4.0), which means that the manuscript, images, and Supporting Information files will be freely available online, and any third party is permitted to access, download, copy, distribute, and use these materials in any way, even commercially, with proper attribution. For more information, see our copyright guidelines: http://journals.plos.org/plosone/s/licenses-and-copyright.

3.1.    You may seek permission from the original copyright holder of Figure 1 to publish the content specifically under the CC BY 4.0 license.

3.2.    If you are unable to obtain permission from the original copyright holder to publish these figures under the CC BY 4.0 license or if the copyright holder’s requirements are incompatible with the CC BY 4.0 license, please either i) remove the figure or ii) supply a replacement figure that complies with the CC BY 4.0 license. Please check copyright information on all replacement figures and update the figure caption with source information. If applicable, please specify in the figure caption text when a figure is similar but not identical to the original image and is therefore for illustrative purposes only.

Reviewers' comments:

Reviewer's Responses to Questions

**Comments to the Author**

1. Is the manuscript technically sound, and do the data support the conclusions?

Reviewer #1: Partly

Reviewer #2: Yes

Reviewer #3: Yes

2. Has the statistical analysis been performed appropriately and rigorously? 

Reviewer #1: I Don't Know

Reviewer #2: N/A

Reviewer #3: Yes

3. Have the authors made all data underlying the findings in their manuscript fully available?

Reviewer #1: Yes

Reviewer #2: Yes

Reviewer #3: Yes

4. Is the manuscript presented in an intelligible fashion and written in standard English?

Reviewer #1: Yes

Reviewer #2: Yes

Reviewer #3: Yes

5. Review Comments to the Author

Reviewer #1: This manuscript reports one experiment in which the preferences for different prestige cues are compared, in the context of a quiz game. The authors conclude that prestige drives the tendency to choose who must be copied during the game. Additionally, domain-specific prestige is preferred to cross-domain and domain-general prestige, whereas domain-general prestige is preferred to cross-domain prestige, thus revealing a hierarchy of prestige sources.

In general, I think that the manuscript has merit. The research question is sound, and the methods seem adequate to address the question. The writing is overall clear, although I had some difficulties in understanding the analyses.

What follows is a list of more specific comments on the article.

-First of all, I do not understand well why the phenomenon studied here (relying on prestige cues to choose who to learn from) is presented as a “bias”. It could certainly be understood as a simple rule, or heuristic. However, as far as I know, the term bias conveys (perhaps implicitly) the idea of systematic error or departure from a normative standard. But, how can we determine the standard (unbiased) behavior in a task like this? Is trusting individuals according to their prestige more biased than using their performance as a cue? In fact, it seems like a reasonable rule that should lead to good decisions most of the time (the Introduction describes the “bias” as “adaptative”). I think this needs further comment and clarification, perhaps in the Introduction.

-I appreciate the transparency in the reporting of data and results (sharing data and code, pre-registering hypothesis).

-Although the article is generally clear, I found it difficult to follow in the Analyses/Results section. I am not sure that the way the manuscript is organized actually helps. On the one hand, presenting the predictions and the analyses schematically, before the results, seemed useful the first time I read it. However, in this study, we have different analyses with different dependent variables (if I understood well: probability of choosing the predicted option, copying rate, and quiz score). When we read the Results section, it is easy to forget which variable we are talking about, or what “prediction 3” (explained before) means. I needed to go back and forth several times to understand what the analyses were actually doing. I suggest to improve this by either organizing the text differently, or by adding explanations so that we do not need to go back and read the predictions/analysis plan. Perhaps organizing this section by dependent variables could help (i.e., first we analyze one variable, and when we finish we start with the next one, and we always include comments to remind the purpose and interpretation of the analysis), but I’m not sure.

-When interpreting Figure 3, I wonder about the usefulness of this analysis, which in addition is quite complex to understand. The dependent variable is the probability of choosing the predicted option. Thus, the higher the model estimates, the better the researchers’ predictions were. This is a transformed variable that is not easy to interpret (note that in one condition, choosing “Domain-general” is coded as 1, but in other conditions the same answer is coded as 0). For example, how should we interpret the lack of differences between conditions in this variable? If you had found a difference in one condition against the others, how would you interpret it? I am not sure that the statistical model that is used here is actually useful. At least, I would need the authors to interpret the results when they are presented so that I can fully understand what the analysis reveals. Otherwise I had the impression that the comparison between conditions is meaningless, but I could be wrong.

-I also found the design a little bit overcomplicated, and this may be due to the procedure used.

First, experiments need manipulation and control of variables. Here, the only manipulated variables are the two options that are presented to each participant. The rest of parameters (the participant’s score, the participants’ prestige...) are uncontrolled, as they are generated spontaneously by participants (do I understand well the procedure?). If this is an experiment and we have clear predictions as to which cues should play a role, why not manipulating them directly? This could be achieved easily. For example, imagine that participants are presented a list of potential players to copy. But instead of real players (i.e., other participants) whose scores and parameters are beyond the control of the experimenter, you show them a list of fictitious players whose parameters are fixed by the experimenter (e.g., Player 111 has been copied 3 times in this domain, 5 times in another domain, etc.). Thus, you could manipulate the variables while controlling for other factors. The consequence of surrendering this degree of control is that the data we obtain are weaker, as we do not know whether the performance of the participants is actually affecting their individual decisions to copy/not copy, etc. What happens if, by chance, your 10 participants are very good at Geography and nobody asks for help? This affects the prestige cues that are crucial to your conclusions. At least a clarification about why the current procedure was preferred and what the benefits are is due.

Second, what is the justification for presenting only two options when copying the answers (e.g., “Domain-Specific vs. Cross-Domain”)? I understand that this simplifies the task of the participant. However, the design becomes more complicated (to carry out, to report and to interpret) as we need to compare between pairs of conditions. Why is it not possible to present a list of options that convey all possibilities (Domain-Specific, Domain-General, Cross-Domain, Random Cue)? If there are good reasons to present only two options, I would like to read them in the manuscript.

-The choice of this procedure/design also has consequences in terms of statistical power. Although the sample size is overall good, we have small cell sizes. This compromises the generalization of the results.

-Additionally, and related to the previous point, I would like to see a justification for the sample size: if it was not decided a priori, a sensitivity analysis would suffice, to have an idea about the smallest effect that your study could detect with reasonable power. Choosing a sample size similar to a previous study is generally not a good strategy to decide the N.

-The authors use a variety of domains to test the generalizability of their results. As the authors discuss (in the Discussion), the correlation between domains that participants assume can be important. Could this be a problem when interpreting the current data? For example, if participants believe that the skills necessary to succeed in two of the tasks are similar, or the same (e.g., both Geography and Language are based on general knowledge, therefore the ones succeeding in Geography should do fine in Language), then they might show apparent preference for general domain prestige. I wonder if the results could be different depending on the set of domains used in the experiment: whether they are closely related or they are completely independent (e.g., a pshysical task and an intellectual task). Perhaps this could be added to the Discussion.

More specific comments:

-I would like to see additional information about what participants see on the screen. For example, when they choose to “ask someone else”, what information is then presented? We have screenshots for the questions, but not for these informative screens.

-Is it possible to name the conditions in a more informative way than “condition A, condition B...”? I had to check the Table multiple times to remember which cues are being contrasted in each condition.

-I think that describing in the paper the function used to compute the Pearson coefficients (“cor.test()”) is unnecessary. Is there anything in the way this function works that is important or unusual so that it deserves comment?

-I am curious as to why the credible intervals have a width of 89%. I get that the popular option of 95% (at least it is more popular in frequentist analyses) means less stable estimations. By then, why not 90%? I don’t mean that the manuscript or the analyses are wrong, but the choice of 89% had me wonder, and there is no justification in the text.

Reviewer #2: Introduction

- It is necessary to provide a complete definition of prestige. In this vein, it would be interesting to clarify if there is any connection between source prestige and source credibility, commonly identified to consist of expertise and trustworthiness (for a review, see Pompitakpan, 2004) or other attributes, such as age or gender.

- Please, provide an explanation of the difference between prestige information and direct success information (line 67).

- Please, justify the reason why it is important to better understand domain-specificity and -generality of prestige-bias. Which kind of implications might these two kinds of prestige bias have in real life? It could be very helpful to add some specific examples.

Method_Procedure

In general, the information provided in this section is incomplete or not clear. In several cases, it is necessary to consult previous studies of the authors to fully understand the content. For instance:

- What is the reason why if everyone chose to “Ask Someone Else” then participants were shown a message saying “sorry, everyone chose to ‘ask someone else’ so no one can score points for this question?

- The step “They then chose a demonstrator whose answer they used for that question (lines 207-208)” is confusing. Could you offer a more detail explanation?

Discussion

- Please, explain in more depth the implications the ‘hierarchy’ of prestige detected might have in real life.

Minor comments

7. Line 470: There is a double comma.

Reviewer #3: This is a well-written manuscript on an interesting topic for evolutionary psychology with sound methods and clean results. The pre-registration is also sound and the protocol is followed carefully by the authors. It is true that probably the results are not smashing but this is not a concern in PLOS One (and not a concern in general for me). Therefore, I can just congratulate the authors. If I am not wrong, this is the first time I recommend acceptance of a manuscript in the first revision round.

I have just one very minor comment: it would be interesting to test whether participants choose to copy those who are most copied when they can choose to copy based on scores; that is, what of these two information cues is preferred when are both present? Unfortunately, as far as I understand, the current design does not allow for this analysis and should be left for future research. Maybe the authors want to discuss this point.

Antonio M. Espín

6. PLOS authors have the option to publish the peer review history of their article (what does this mean?). If published, this will include your full peer review and any attached files.

Reviewer #1: No

Reviewer #2: No

Reviewer #3: **Yes: **Antonio M. Espín

---

## [Author Response · Author response to Decision Letter 0]

31 Mar 2021

Thank you for taking the time to carefully read our manuscript. We hope our revisions have now made the methods and results section clearer. We’ve also altered the reporting of the correlation coefficients, as well as other substantial clarifications required by the reviewers, included in the attached "response to reviewers" file.

---

## [Decision Letter · Decision Letter 1]

18 May 2021

PONE-D-20-39514R1

Trusting the experts: the domain-specificity of prestige-biased social learning

PLOS ONE

Dear Dr. Brand,

Thank you for submitting your manuscript to PLOS ONE. After careful consideration, we feel that it has merit but does not fully meet PLOS ONE’s publication criteria as it currently stands. Therefore, we invite you to submit a revised version of the manuscript that addresses the points raised during the review process.

Below you will find the comments provided by Reviewer 2 and a new reviewer (4). Reviewer 1 declined the invitation to read your manuscript again. As you can see, Reviewers 2 and 4 make radically different recommendations: while Reviewer 2 now recommends acceptance, Reviewer 4 recommends outright rejection. My own impression is that the concerns raised by Reviewer 4 (partially overlapping with those of Reviewer 1) do obscure the interpretation of the results and merit, at least, a detailed discussion in the main text. In the same vein, I think that some of the problems previously detected by Reviewer 1 should lead to further changes in the text.

Both Reviewers 1 and 4 note that the fact that you didn't manipulate prestige cues experimentally undermines your interpretation of the results. This is perhaps clearest in the new review provided by Reviewer 4. The main concern is that because during Round 1 participants only had information about the domain-specific score of the other participants, then all measures of prestige in Round 2 can be seen as proxies for the domain-specific score of the other participants. That is, the information that participants see in Round 2 about who was imitated most often in Round 1 must be determined by the domain-specific scores in Round 1. This obscures whether the results seen in Round 2 are actually driven by prestige cues or rather by inferred domain-specific accuracy. In your response to Reviewer 1 you argue that your procedure provides a more natural test of how prestige dynamics arise in the real world. This is a fair point, but I think it deserves an explicit discussion in the main text. It is absolutely fine to emphasize the advantages of using a naturalistic procedure, but the reader should also be alerted about the potential problems of this strategy.

If I understand the design properly, condition D was not part of the original preregistered protocol and was added after data were already collected for conditions A-C. An unfortunate consequence of this is that all the analyses that include condition D fall outside the scope of the pre-registered protocol. Please note this in the main text (i.e., that all models involving condition D depart from the protocol). Also, condition D is introduced in Table 1, but the fact that this condition was not part of the preregistered protocol is not explained until line 450. Please, explain as early as possible in the ms that condition D was not part of the preregistration.

I understand that conditions A-D were manipulated between participants, but I don't think this is clearly stated anywhere in the manuscript. Please, say so explicitly and consider referring to Groups A-D instead of conditions A-D. Note that this is also relevant to understand Reviewer 4's concerns about your mixed models (i.e., whether participants are nested within groups or crossed).

Reviewer 1 complained that the general procedure was difficult to follow and suggested adding a figure. I think that the new figure 1 does help the reader understand the procedure, but it would be even better to present not only a trial from one particular condition in Round 2, but examples also from Round 1 and the remaining conditions in Round 2.

Reviewer 1 also questioned the adequacy of Figure 3. I do share the feeling that the information conveyed by Figures 2 and 3 is somewhat redundant.

Reviewer 1 asked for some justification of sample size. Although your response is compeling it is true that in the present version of the ms it is difficult to appreciate whether your sample is sufficiently sensitive given the effects you are studying. I don't think a power analysis would be appropriate here (because you are not using frequentist stats), but it would be nice to have some measure that allowed the reader to infer whether the number of observations is large enough to conclude with certainty that the effects you find are conclusively different from zero (or not). I think that Bayes factors against the null hypotheses would serve this purpose well.

Reviewer 1 complained that the analytic strategy was also difficult to follow. I do think that this version is clearer but it would be even better to merge completely the paragraphs where you explain the predictions and the paragraphs where you explain how you will test them. In other words, explain on lines 380-385 that you will test prediction 1 using correlation coefficients. Merge lines 395-401 with Prediction 2 and in Prediction 3 simply explain that the analysis will be as in Prediction 3; and follow the same logis with the remaining predictions and analysis plans. Otherwise the reader is constantly forced to go back to each prediction when reading the proposed analyses.

Minor comments:

line 60: "Here, 'bias' in meant..." -> "in" should be "is"

line 141: Double space after "four"

lines 460-472: on a first read, it is unclear whether this refers to all the analyses or just to the one immediately above. Please clarify this here.

table 2: please report confidence intervals (or credibility intervals) for these correlation coefficients, so that the reader can appreciate to what extent the numerical differences between them are meaningful or not.

Starting on line 568 you report for each the model intercepts and refer to them as "mean coefficient estimate". The reader is forced to go back to the model description to understand that this is an intercept. Please, refer to this as mean intercept estimate instead or alert the reader somehow what these numbers mean.

lines 695-697: The second part of the sentence (i.e., "... and emerge either through unconscious associative learning, conscious deliberation, or both") doesn't add information beyond what's already said in the first half. It simply says that the nature of this type of learning is unknown.

We look forward to receiving your revised manuscript.

Kind regards,

Miguel A. Vadillo, Ph.D.

Academic Editor

PLOS ONE

Reviewers' comments:

Reviewer's Responses to Questions

**Comments to the Author**

1. If the authors have adequately addressed your comments raised in a previous round of review and you feel that this manuscript is now acceptable for publication, you may indicate that here to bypass the “Comments to the Author” section, enter your conflict of interest statement in the “Confidential to Editor” section, and submit your "Accept" recommendation.

Reviewer #2: All comments have been addressed

Reviewer #4: (No Response)

2. Is the manuscript technically sound, and do the data support the conclusions?

Reviewer #2: Yes

Reviewer #4: Partly

3. Has the statistical analysis been performed appropriately and rigorously? 

Reviewer #2: I Don't Know

Reviewer #4: I Don't Know

4. Have the authors made all data underlying the findings in their manuscript fully available?

Reviewer #2: Yes

Reviewer #4: Yes

5. Is the manuscript presented in an intelligible fashion and written in standard English?

Reviewer #2: Yes

Reviewer #4: Yes

6. Review Comments to the Author

Reviewer #2: The authors of the manuscript "Trusting the experts: the domain-specificity of prestige-biased social learning" have successfully answered the comments I raised in the previous version.

I would still try to make clearer the relationship, if any, between prestige and expertise but this is only a suggestion that in no way modifies my position to accept the manuscript as it stands.

Reviewer #4: In the experiment reported in this manuscript, participants were given the opportunity to choose to answer each one of 100 quiz questions (in two rounds) by themselves, or copying from another participant. In the different conditions of the experiment, the available sources to copy from in the second round of questions were manipulated (using different pairs of sources in each condition) in such a way that participants could choose between two sources with different types of prestige (within-field, between-fields, general, or a random cue). The authors conclude that any type of prestige cue drives decisions (is preferred) relative to no prestige, within-field relative to general and between-fields, and general to between-fields.

As a general comment, I commend the authors for the transparency in performing and reporting the study. Preregistration, and sharing data and code, as done here, is a warranty that the authors are not using analysis flexibility to surreptitiously manipulate results in their favor. Moreover, the theoretically relevant effects are probably strong enough to provide substantial evidence in favor of all the main hypotheses, regardless of the statistical approach used.

Still, the preregistered plan describes the model to test main hypothesis as a Bayesian GLME with the following structure: Chose_predicted ~ intercept + 1|condition + 1|Participant + 1|group +1|topic

If this is the model that was finally run (I am sorry I am not familiar with the specific syntax of the package used for Bayesian analysis here), why is condition modelled as a random-effects factor (random intercept)? As far as I know, random-effects factors are those for which levels can be considered as randomly sampled from the set of possible ones, whereas here levels are actively manipulated. And also, this syntax suggests that group and participant factors were crossed, but in the design participants are actually nested in groups. Please clarify these issues, or correct me if I am misinterpreting something.

My most important concern, however, is not methodological but conceptual. The authors claim that prestige emerges from the task, but I am afraid the very concept of prestige is totally unnecessary to account for the results.

As detailed in the manuscript (and in the authors’ response to a reviewer from the previous round), participants had access to the in-field accuracy of answers from the other participants in the first round. It is true that participants do not have access to information on whether peers also tended to choose the responder with the highest number of correct answers, but assuming that “most people will do as I do” seems rather straightforward to me. In other words, the task is not letting prestige “emerge”, it is just providing an almost perfect proxy (the most frequently chosen responder) to in-field objective accuracy.

Consequently, if, as a responder, I have a perfect proxy to in-field accuracy I also have a less perfect proxy to general accuracy (as in-field accuracy mathematically contributes to general-accuracy), and an even less perfect proxy to between-fields accuracy. That is, the ordering of preferences in the second round does not require to assume the use of any prestige cue at all, but just plain reasoning based on estimated accuracy.

At the present moment, I see no necessity for prestige as an intermediate explanatory construct. As mentioned by one of the reviewers in the previous round, it would have been more convincing to actively manipulate prestige cues. Hence, unless I am wrong here (and I hope I am, given the carefulness with which this study has been carried out), my recommendation will be not accepting this manuscript for publication.

7. PLOS authors have the option to publish the peer review history of their article (what does this mean?). If published, this will include your full peer review and any attached files.

Reviewer #2: No

Reviewer #4: No

---

## [Author Response · Author response to Decision Letter 1]

2 Jul 2021

Our detailed reviewer response document can be found in the submission files, along with the cover letter.

---

## [Editor Report · Decision Letter 2]

9 Jul 2021

PONE-D-20-39514R2

Trusting the experts: the domain-specificity of prestige-biased social learning

PLOS ONE

Dear Dr. Brand,

Thank you for submitting your manuscript to PLOS ONE. After careful consideration, we feel that it has merit but does not fully meet PLOS ONE’s publication criteria as it currently stands. Therefore, we invite you to submit a revised version of the manuscript that addresses the points raised during the review process.

I think that the ms is essentially ready for publication. Before accepting it, I'd like the authors to consider the following changes:

lines 60-62 "Crucially, prestige-bias only evolved, and is only adaptive, because the prestige was first acquired due to success..." -> This sounds like a proven fact ("only evolved") but it is actually speculation based on cultural evolutionary theory of prestige. Consider changing the beginning of the sentence to "According to the cultural evolutionary theory of prestige..." and then remove "Consistent with this cultural evolutionary theory of prestige" in the following sentence.

lines 146-148 In "As before, we use participants..." consider rewriting as "As before, instead of manipulating prestige experimentally, we use participants..."

We look forward to receiving your revised manuscript.

Kind regards,

Miguel A. Vadillo, Ph.D.

Academic Editor

PLOS ONE
---

## [Author Response · Author response to Decision Letter 2]

12 Jul 2021

Thank you for taking the time to constructively engage with our work. We have happily made the final changes suggested, as can be seen in our tracked changes document, and the "response to reviewers" document.

---

## [Editor Report · Decision Letter 3]

15 Jul 2021

Trusting the experts: the domain-specificity of prestige-biased social learning

PONE-D-20-39514R3

Dear Dr. Brand,

We’re pleased to inform you that your manuscript has been judged scientifically suitable for publication and will be formally accepted for publication once it meets all outstanding technical requirements.

Kind regards,

Miguel A. Vadillo, Ph.D.

Academic Editor

PLOS ONE
---

## [Editor Report · Acceptance letter]

29 Jul 2021

PONE-D-20-39514R3 

Trusting the experts: the domain-specificity of prestige-biased social learning 

Dear Dr. Brand:

I'm pleased to inform you that your manuscript has been deemed suitable for publication in PLOS ONE. Congratulations! Your manuscript is now with our production department. 

Kind regards, 

on behalf of

Dr. Miguel A. Vadillo 

Academic Editor

PLOS ONE